# On a new robust workflow for the statistical and spatial analysis of fracture data collected with scanlines (or the importance of stationarity)

Andrea Bistacchi[1], Silvia Mittempergher[2], Mattia Martinelli[1], Fabrizio Storti[3]

[1]Dipartimento di Scienze dell'Ambiente e della Terra, Università degli Studi di Milano Bicocca, Piazza della Scienza, 4, 20126 Milano

[2]Dipartimento di Scienze Chimiche e Geologiche, Università degli Studi di Modena e Reggio Emilia, Via G. Campi 106, 41125 Modena

[2]NEXT - Natural and Experimental Tectonics Research Group, Dipartimento di Scienze Chimiche, della Vita e della Sostenibilità Ambientale, Università degli Studi di Parma, Parco Area delle Scienze 11/a, 43124 Parma

*Correspondence to*: Andrea Bistacchi (andrea.bistacchi@unimib.it)

**Abstract.** We present an innovative workflow for the statistical analysis of fracture data collected along scanlines, composed of two major stages, each one with alternative options. A prerequisite in our analysis is the assessment of stationarity of the dataset, that is motivated by statistical and geological motivations. Calculating statistics on non-stationary data can be statistically meaningless, and moreover the normalization and/or sub-setting approach that we discuss here can greatly improve our understanding of geological deformation processes. Our methodology is based on performing non-parametric statistical tests, that allow detecting important features of the spatial distribution of fractures, and on the analysis of the cumulative spacing function (CSF) and cumulative spacing derivative (CSD), that allows defining the boundaries of stationary domains in an objective way. Once stationarity has been analysed, other statistical methods already known in literature can be applied. Here we discuss in details methods aimed at understanding the degree of saturation of fracture systems based on the type of spacing distribution, and we evidence their limits in cases where they are not supported by a proper spatial statistics analysis.

## 1 Introduction

The analysis of fracture systems is a traditional topic in structural geology and rock mechanics (e.g. Pollard and Aydin, 1988, Twiss and Moores, 2006), and it has been recognized from a long time that the quantitative characterization of fracture networks is necessary both in fundamental studies aimed at understanding brittle deformation processes in different geological environments (e.g. Jaeger et al., 2007; Scholz, 2019; Schultz, 2019), and in applications such as engineering rock mechanics (e.g. Hoek, 1980) and the characterization and modelling of subsurface fluid flow (e.g. Gleeson and Ingebritsen, 2012).

For "fractures" in this contribution we use the broad definition given e.g. by Twiss and Moores (2006), Davis et al. (2012), or Schultz (2019), who include in "fractures" or "cracks" all kinds of brittle discontinuities, such as joints, veins, shear fractures, (micro-faults) and in some cases even stylolites and other "anti-cracks". Following the same Authors, a fracture set is defined

as a cogenetic set of fractures showing the same kinematics and orientation (with some variability), while a fracture system or network includes all fracture sets that are present in a volume of rocks.

For years, the common way of collecting quantitative fracture data has been based on scanlines: drawing a line, or laying a
tape measure, on an outcrop and collecting all intersections between this line and fractures (e.g. Terzaghi, 1965; Hobbs, 1967; Priest and Hudson, 1981). Equivalent 1D datasets are also obtained from boreholes, both from direct observation of cores and/or from geophysical imaging methods (e.g. Prioul and Jocker, 2009). The main advantage of scanline surveys is that they can be easily carried out in relatively short times and in a multitude of outcrops, hence this technique has become a standard, also in engineering applications (ISRM, 1978).

The minimum goal of scanline surveys is the measurement of fracture spacing - the distance between two neighbouring fractures, and the analysis of derived parameters such as fracture density (number of fractures per unit length) or intercept (inverse of density, e.g. Priest and Hudson, 1981).

Another fundamental information that can be obtained from scanlines is the 1D spatial distribution (e.g. Swan and Sandilands, 1995; Laubach et al., 2018), i.e. whether fractures are randomly distributed in space, or they show some form or spatial
organization, defined as any form of departure from complete randomness (Figure 1a). We can consider two main and opposite types of organization: (i) regular, uniform, periodic arrangements (Figure 1b), and (ii) arrangements with some form of clustering (Figure 1c) or repeating pattern (Figure 1d)(e.g. Swan and Sandilands, 1995; Bonnet et al., 2001; Marrett et al., 2018).

The spatial organization of fractures can be interpreted in function of the processes responsible of their formation and evolution.
Fractures nucleate at randomly distributed weakness points in the host rock (e.g. Schultz, 2019) and then propagate, increasing their length, initially without interacting with each other (e.g. Spyropoulos et al., 2002). When the number of fractures and their dimensions increase (e.g. fracture density and intensity increase), fractures start interacting, with two possibly opposite effects (e.g. Scholz, 2019). If the material is effectively weakened by fractures (e.g. at fracture tips where stress concentrations occur or due to diffuse damage in the material), fractures tend to propagate one towards each other, or to nucleate one close to
each other, in an "attractive" process (e.g. Crider and Pollard, 1998). When instead stress shadows develop, fracture nucleation or propagation is inhibited close to other fractures, in a "repulsive" process (e.g. Spyropoulos et al., 1999).

Mechanical and statistical simulations have shown that repulsive processes result in regular/uniform/periodic spatial distributions where the mean spacing might reflect the width of stress shadows (Rives et al., 1992; Bai and Pollard, 2000; Tan et al., 2014). On the other hand, attractive processes result in clustering of fractures or in the development or repeating patterns,
that sometimes, but not always, show fractal distributions (e.g. Gillespie et al., 1993; Turcotte, 1997; Marrett et al., 2018). As shown by Olson (2004), also subcritical crack growth can result in clustered fracture distributions, that can be described as fracture corridors (*sensu* Sanderson and Peacock, 2019).

A different type of spatial organization, depending from external tectonic and/or lithological controls, is represented by large-scale trends in fracture density and spacing (Figure 1e) that can develop e.g. in fault damage zones, with fractures that are
more densely spaced close to the fault core and more dispersed as the distance increases (e.g. Vermilye and Scholz, 1998;

Faulkner et al., 2008; Bistacchi et al., 2010; Choi et al., 2016), or when fracturing is controlled by folding (Tavani et al., 2006; Cosgrove, 2015; Tavani et al., 2015).

Following the fundamental paper by Hobbs (1967), many authors investigated the relationships between the spatial distribution of fractures and the statistical distribution of their spacing (e.g. Rives et al., 1992; Gross, 1993; Tan et al., 2014). These relationships are based on sound statistical ground. For instance, if we consider a Poisson distribution in 1D (a perfectly random distribution of events along a line), the spacing between neighbouring events will be characterized by a negative exponential distribution, with numerically equal mean and standard deviation (Swan and Sandilands, 1995). If we consider a regular spatial distribution, the mean spacing will be sharply defined, the spacing standard deviation will be much smaller, and the statistical distribution of spacing will be close to a symmetrical normal distribution (Rives et al., 1992). Intermediate situations will show spacing distributions with intermediate skewness, such as Gamma or log-normal distributions (Rives et al., 1992).

These observations lead many Authors using the distribution of spacing to characterize the spatial distribution, as if the above-mentioned relationships can be taken as *biunivocal* relationships. For instance, Gillespie et al. (1999) proposed a method to characterize spatial distribution based on the coefficient of variation $Cv = std\ deviation/mean$ that is supposed to be $Cv > 1$ for clustered systems, $Cv = 1$ for perfectly random systems (since $std\ deviation = mean$ in a negative exponential distribution), and $Cv < 1$ for systems with a regular/uniform spatial distribution.

However, for instance Wang et al. (2019) warned against using this methodology that is not able to "capture mixtures of highly clustered and more regularly spaced patterns". We will demonstrate in the following that the problem is more general and lies in the fact that the relationship between spatial distribution and the distribution of spacing is *not always biunivocal*. A more robust approach to detect spatial organization was recently undertaken by Marrett et al. (2018), who proposed the normalized correlation count $NCC$ as a useful tool to detect spatial organization that departs from complete randomness ($NCC = 1$), as clustering ($NCC > 1$) or regular spacing, seen as anti-clustering ($NCC < 1$). One major advantage of this approach is that it is possible to detect how spatial organization changes as a function of wavelength, for instance in cases where fractures are distributed in clusters, and clusters have a regular spacing, but the distribution of fractures within a single cluster is fractal (as e.g. in Wang et al., 2019).

A simple yet promising approach for studying spatial organization is based on cumulative distributions. Gillespie et al. (1999) plotted the cumulative vein thickness vs. vein position along a scanline to characterize spatial distribution. Choi et al. (2016) plotted the cumulative number of fractures along scanlines and uses marked changes in slope of this curve to reveal fault zone architecture features, such as the boundary of the damage zone. However, both Authors did not develop quantitative statistical criteria in their analysis.

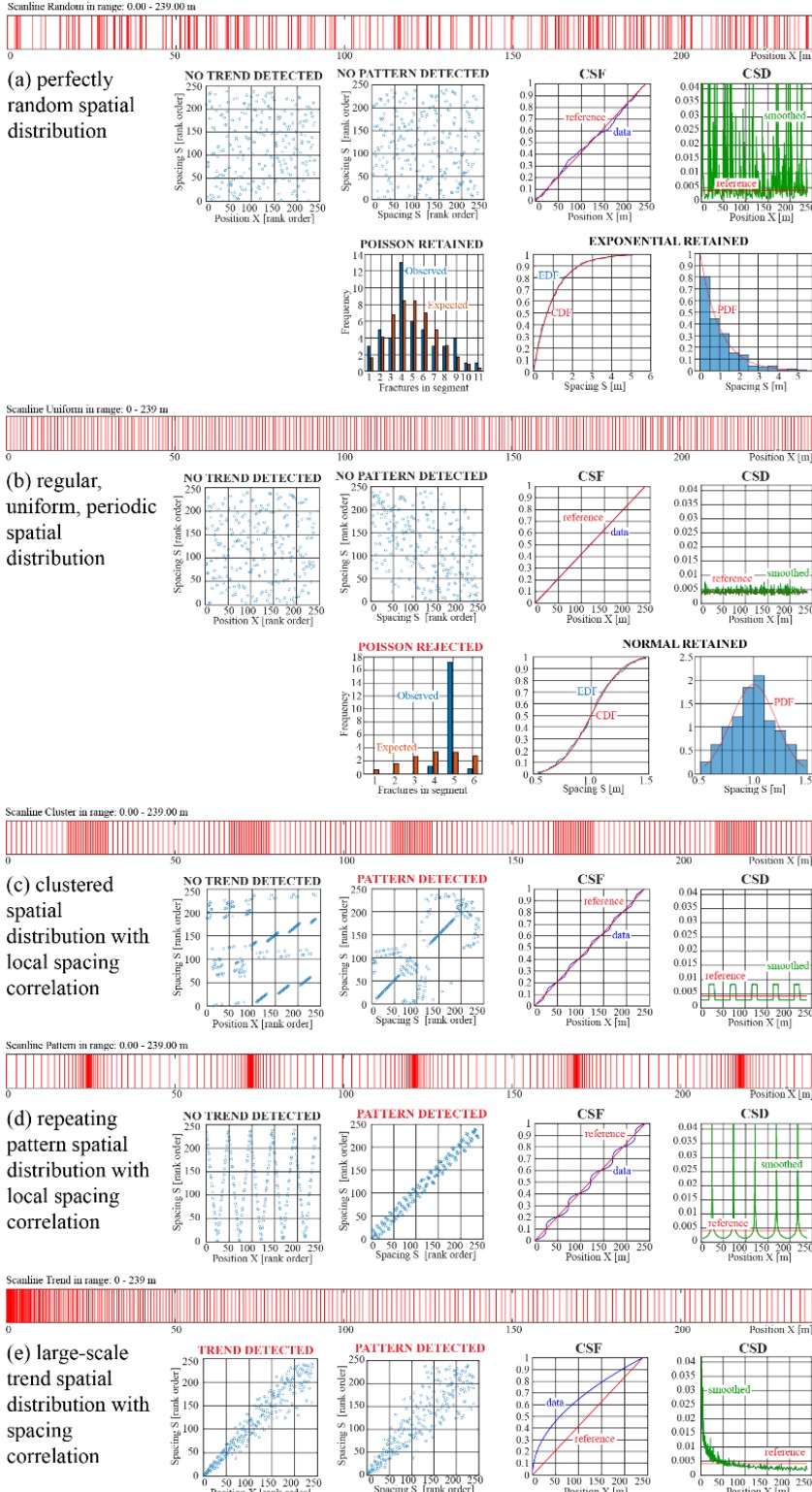

(a) perfectly random spatial distribution

(b) regular, uniform, periodic spatial distribution

(c) clustered spatial distribution with local spacing correlation

(d) repeating pattern spatial distribution with local spacing correlation

(e) large-scale trend spatial distribution with spacing correlation

**Figure 1** Schematic examples of different spatial distributions of fractures sampled along a scanline or borehole. In all examples fracture density is the same. For each example we show, where applicable: barcode plot of all fracture intersections along the scanline; rank order position vs. rank order spacing plot; rank order spacing at i-th position vs. rank order spacing at i+1-th position; data Cumulative Spacing Function (CSF - blue) compared with reference CSF (red); data Cumulative Spacing Derivative (CSD -blue) compared with reference CSD (red), and CSD smoothed with moving median filter (green); comparison of observed vs expected frequencies for a Poisson random distribution; best-fit spacing cumulative distribution function (CDF) compared with empirical distribution function (EDF); best fit spacing probability density function (PDF) compared with histogram.

In this contribution we will first focus on a rigorous framework to measure spatial and spacing distributions in scanlines, on the relationships between these distributions, and particularly on the debated possibility to infer spatial distribution from spacing distribution (is it a biunivocal relationship?). An important point, not addressed by previous Authors, will be dedicated to defining stationarity of a statistical sample within a given spatial sampling domain – a prerequisite to detect any meaningful statistics of the sample itself, and to extrapolate it to the underlying population. We will then propose non-parametric and parametric statistical tests that can be used to characterize the distributions, and we will propose a rigorous workflow that can be applied to the practical analysis of scanline data.

## 2 Rationale

### 2.1 Fracture position and spacing in the scanline reference frame

To completely characterize the spatial distribution of fractures and their spacing, we need to define two stochastic variables: position and spacing. Having collected $N$ fracture intersections along a scanline of length $L$, chosen perpendicular to the fracture set's mean plane (Priest and Hudson, 1981), we will call position $X_i^F$ the distance between the origin of the scanline reference frame and the $i - th$ fracture intersection (Figure 2). Spacing is defined as the distance between one fracture at position $X_i^F$ and the next fracture at position $X_{i+1}^F$, so it is a property of the position of both fractures. Under a different point of view, spacing can be also seen as the length of the block of intact rock between two fractures. For this reason, we calculate the $i - th$ spacing $S_i$ at the $i - th$ "block" position $X_i^B$ as follows (Figure 2):

$$X_i^B = \frac{X_i^F + X_{i+1}^F}{2}$$

$$S_i = X_{i+1}^F - X_i^F$$

We will see in Sect. 2.4 that associating each spacing measurement to its position is fundamental to study the spatial distribution of fractures in terms of the spatial distribution of their spacing.

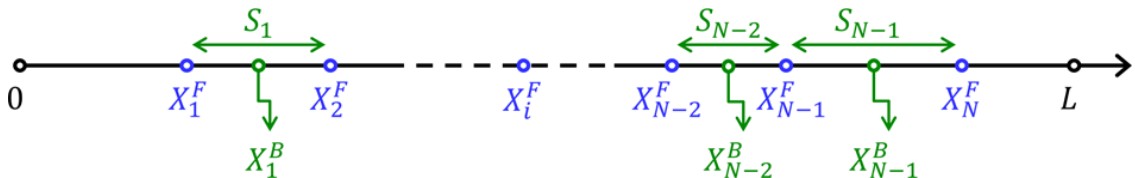

**Figure 2 Scanline reference frame with definition of position and spacing variables.** $L$: scanline length; $X_i^F$: position of $i-th$ fracture intersection; $S_i$: spacing of $i-th$ "block" with mid-point position $X_i^B$.

If local outcrop or borehole conditions do not allow collecting the scanline perpendicular to the mean plane of the fracture set, we will apply the Terzaghi (1965) correction to obtain the true position from the apparent one:

$$X_{TRUE} = \frac{X_{APPARENT}}{cos(\alpha)}$$

where $\alpha$ is the angle of deviation between the actual scanline and optimal one (perpendicular to the fracture set mean plane). Spacings will be automatically correct if calculated from true positions. Fracture positions can be schematically represented

as "barcode plots" (Figure 1), useful for a visual inspection of the spatial distribution of fractures.

## 2.2 Stationarity as a prerequisite for statistical analysis

In statistics, a stationary process is a stochastic process whose characteristic probability distributions do not change when the domain of the analysis (e.g. in space or time) is varied (e.g. shifted, shrink, enlarged; e.g. Wasserman, 2004). Stationarity is a prerequisite in many kinds of analyses performed on time series, and it is a fundamental concept also for spatially distributed

variables. In geostatistical studies on spatially distributed variables, a variable is said to be stationary if there is no significant drift or trend within a specified spatial domain (distance, area, or volume), where drift or trend is defined as the component of a regionalized variable resulting from large-scale processes that can be defined with deterministic analytical functions of the spatial variables (Swan and Sandilands, 1995; Borradaile, 2003). An example of trend in fracture studies is the exponential decay in fracture density that can be found in some fault damage zones (e.g. Caine and Forster, 1999; Mitchell and Faulkner,

2009). Within this framework, residuals represent the completely random component of the regionalized variable, that, if a trend is present, can be obtained by normalizing the regional variable with the trend (Swan and Sandilands, 1995).

Even if this analysis is not routinely performed, understanding if our dataset is affected by a trend (Figure 1e) is fundamental in every kind of statistical analysis on spatial variables since, if a trend is present, all statistics will be affected by the choice of the sampling domain. If for instance we calculate the sample mean of fracture spacing in a fault damage zone showing an

exponential trend, the mean will change depending on the position and length of the scanline. Fracture spacing is not stationary in this case and its mean is meaningless. The same happens, at a smaller scale (e.g. a small segment of the scanline), if fractures are clustered (Figure 1c) or arranged in a pattern (Figure 1d).

According to the general definition given above, a stationary process is a process whose statistics do not change when the sample is changed, moved, shifted, or resized in space (or time etc.; Wasserman, 2004). With reference to Figure 1, this

restrictive condition is met only in cases when fractures are randomly distributed (Figure 1a), or regularly spaced (Figure 1b).

## 2.3 Non-parametric correlation

Non-parametric statistical methods discussed here are those in which (Davis, 2002) (i) data are represented with ordinal or rank-order scales instead of interval- or ratio-scales and (ii) data are not required to fit parametric statistical distributions (e.g. the normal or exponential distributions). Since they are independent from any assumption or statistical model, they are quite useful, particularly in the first phases of analysis (Swan and Sandilands, 1995).

The non-parametric Spearman's rank correlation coefficient measures the correlation between two rank-order variables. These are variables obtained by sorting numerical data (interval- or ratio-scale variables) from the smaller to the larger value, and then replacing each value with an integer representing its position in the sequence. If we take a dataset with $N$ data pairs $x_i$ and $y_i$, and the rank-order variables are $R(x_i)$ and $R(y_i)$, the Spearman's correlation coefficient $R_S$ is given by:

$$R_S = 1 - \frac{6 \sum_{i=1}^{N} \big(R(x_i) - R(y_i)\big)^2}{N(N^2 - 1)}$$

For instance, in case of a perfect correlation between $x_i$ and $y_i$ in terms of rank order, we will have, for every $i - th$ data pair, $R(y_i) = R(y_i)$, hence $R_S = 1$.

We can use the Spearman's rank correlation coefficient to test the null hypothesis of no correlation vs. the alternative hypothesis of correlation, considering critical values of $R_S$ or the associated $p - values$ (Wasserman, 2004), so for instance we will reject the null hypothesis at 5% significance if $p - value < 0.05$.

The advantage of the Spearman's rank correlation coefficient with respect to its parametric counterpart  - the Pearson's correlation coefficient – is graphically explained in Figure 3, were we see that non-parametric correlation is more robust in case of outliers, and is also able to detect non-linear correlation, in addition to standard linear correlation (Swan and Sandilands, 1995).

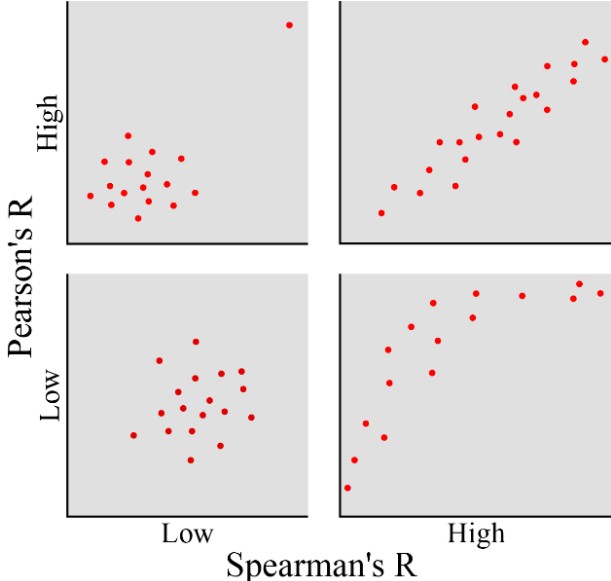

**Figure 3 Comparison of the Spearman's rank correlation coefficient with respect to the parametric Pearson's correlation coefficient. Non-parametric Spearman's correlation is more robust in case of outliers, and is also able to detect non-linear correlation, in addition to standard linear correlation (Swan and Sandilands, 1995).**

### 2.4 Cumulative spacing spatial distribution: CSF and CSD

We have found that, to study the spatial distribution of spacing, it is very useful to plot the cumulative function of spacing, normalized by scanline length, and its first derivative (Figure 1). This approach can be seen as a development of those proposed by Gillespie et al. (1999), Choi et al. (2016), and Sanderson and Peacock (2019), with the important difference that we have developed a new quantitative and objective method to detect segments of a scanline that show a stationary behaviour.

The cumulative spacing function (CSF) corresponds to a plot of the relative proportion of scanline length associated to each "block" limited by a pair of fractures. The reference CSF, associated to a perfectly uniform spatial distribution of fractures, corresponds to a constant-slope line with $slope = 1/scanline\ length$ (Figure 1b). Fracture clusters or other scanline segments with higher-than-average fracture density appear as segments with slope higher than the reference CSF, and the opposite applies to segments with lower-than-average fracture density (Figure 1c-e). The CSF for a random distribution shows just some noise, symmetrically distributed on both sides of the reference CSF (Figure 1a).

The first derivative of the CSF (i.e. the CSD, cumulative spacing derivative) follows the same behaviour. The reference CSD for a perfectly uniform distribution plots as a horizontal line with $height = 1/scanline\ length$ (Figure 1b). Clusters or high-fracture-density scanline segments plot above the reference CSD and lower-fracture-density segments plot below the reference CSD (Figure 1c-e), hence they can be easily recognized and classified with an objective criterion.

By comparing the CSF and CSD, it is possible to define stationary scanline segments, that show a constant slope and hence a constant fracture density. If more than one stationary segment occurs in a scanline, the boundary between different segments is found by comparing the data CSD with the reference CSD. The only difficulty with this approach is that the CSD of natural datasets can be very noisy, with many spikes (see the random distribution in Figure 1a). This problem is solved by smoothing the derivative with a $n - th$ order moving median filter (where $n$ is the kernel size). To avoid biases due to subjective choices on the smoothing kernel size, we propose to retain subset boundaries that are consistently defined when changing the smoothing kernel size over a wide range.

## 3 Our workflow for scanline analysis

Our workflow for the analysis of fracture data collected on scanlines can be subdivided in two major stages. First, we use non-parametric statistics to check whether the dataset is stationary, and, if it is not stationary, we create subsets, and/or we rescale or normalize the data, in order to obtain stationary sets or subsets. Then, we analyse the data (sub-)sets with parametric statistics, following well-established methods to characterize the type of spatial distribution and its parameters.

A web-app, implemented on a MATLAB Web App Server™, is available to perform all the analysis at the URL https://tsgcode.disat.unimib.it/webapps/home/session.html?app=DomStudioFracStat1D.

### 3.1 Stationarity: non-parametric tests, rescaling, and sub-setting

We apply the Spearman's rank correlation coefficient to test the correlation between two pairs of rank variables, the first designed to detect large-scale trends and the other one better suited to detect distributions with clusters or patterns. Our usage was inspired by Swan and Sandilands (1995), with some original variations and developments.

To test the data for large-scale trends (Figure 1e), we test the correlation between the position of each block bounded by two fractures $X_i^B$ and the corresponding spacing $S_i$. When both variables are expressed as rank order, the position rank order is simply $R(X_i^B) \equiv i$ (i.e the block closer to the origin has rank 1, the second 2, etc.), so the correlation coefficient can be written as (for $N$ "blocks"):

$$R_S^{TREND} = 1 - \frac{6 \sum_{i=1}^{N}\left(i - R(S_i)\right)^2}{N(N^2 - 1)}$$

and we compare $R_S^{TREND}$ with tabulated critical values or use the associated $p - value$ to test the null hypothesis of no trend in spacing. We will obtain $R_S^{TREND} = 1$ in case fracture spacing keeps growing steadily and $R(S_i) = i$ for every "block". On the other hand, for a completely random distribution we will obtain $R_S^{TREND} \rightarrow 0$. Plots of the large-scale position-spacing correlation are shown in Figure 1 for all spatial distributions.

Even if large-scale trends are not detected, local small-scale trends could reveal clustering (Figure 1c) or repeating patterns (Figure 1d). We therefore compare spacing between pairs of fractures taken in a sequence along the scanline, using this formulation for the correlation coefficient:

$$R_S^{LOCAL} = 1 - \frac{6 \sum_{i=1}^{N}\left(R(S_i) - R(S_{i+1})\right)^2}{N(N^2 - 1)}$$

and we compare $R_S^{LOCAL}$ with tabulated critical values or use the associated $p-value$ to test the null hypothesis of no local correlation in spacing. In this case $R_S^{LOCAL} \rightarrow 1$ if $R(S_i) \approx R(S_{i+1})$ for a large number of "blocks", and $R_S^{LOCAL} \rightarrow 0$ otherwise. Plots of the local spacing correlation are shown in Figure 1 for all spatial distributions.

The spacing/spacing correlation test used to detect local clustering/pattern must be performed after the position/spacing test used to detect large-scale trend, because a large-scale trend is seen by the local test as a single large-scale cluster, hence the second test alone is not discriminant for local vs. large-scale correlation in spacing.

If the null hypotheses of no large-scale trend and no local clustering/pattern are retained, we can move on to the parametric analysis stage (next section). However, if this is not the case, the dataset can be rescaled/normalized and/or segmented in

subsets to obtain stationary sets.

Rescaling/normalization is best suited when we detect a smooth deterministic trend in spacing, i.e. the fracture density varies continuously along the scanline. In this case we must define the deterministic function of the trend, normalize the data with this function, and the residuals can be considered as a stationary set that can be analysed with methods presented in the next section. The methods to find the deterministic trend function are various, and we feel that they must be guided also by

geological and tectonic observations, so we think that there is no "general method" to complete this task. We will see an example in the first case study (Sect. 4.1), where we study the variable spacing of joints in a laterally tapering layer of sandstone.

Sub-setting is very useful when we observe a stair-stepping pattern, e.g. where we observe lower fracture density domains intercalated by fracture corridors showing markedly higher fracture intensity, or in some fault damage zones (e.g. case study

2, or Martinelli et al., 2020). These segments can be recognized and objectively classified by observing the CSF and CSD, and particularly by comparing the data curves to the reference ones. In our workflow, we (i) select subset scanline segments (i.e. sub-scanlines), generally bounded by points of intersection of the data CSD with the reference CSD, corresponding to changes of slope of the data CSF, then (ii) we test once again each subset for stationarity and, if the result is positive, (iii) we pass it to the parametric analysis stage (next section).

We also tested an alternative test for randomness, by comparing the spatial distribution of $N$ fractures in terms of position $X_i^F$ along a scanline of length $L$ with a theoretical Poisson distribution with density parameter equal to fracture density $\mu_D = N/L$. The discrete Poisson distribution expresses the probability Pr of having $n$ discrete events (fractures), distributed at random in sub-segments of length $l$, as (Swan and Sandilands, 1995):

$$\text{Pr}(n \text{ events in segment } l) = \frac{(\mu_D l)^n e^{-(\mu_D l)}}{n!}$$

The comparison of empirical and model distributions is performed with a $\chi^2$ goodness-of-fit (GOF) test, used to compare the observed $O_j$ and expected $E_j$ frequencies in segments of the scanline, as in Figure 1a-b. To have reliable results, the expected frequency in each class must be $E_j \geq 5$, hence segments are automatically pooled when they show lower values (Swan and

Sandilands, 1995). Unfortunately, this means that this test is reliable only for very large datasets, and even in this case, in our experience, it shows a limited sensibility in detecting random vs. other arrangements.

## 3.2 Evaluation of a parametric distributions of spacing

Once we have segments of scanline that show a stationary behaviour, we can use standard parametric methods to define the type of spacing distribution (e.g. normal vs. exponential) and fit its parameters (e.g. population mean and standard deviation) to the sample fracture spacing data. Then, following a rich literature (e.g. Rives et al., 1992; Tan et al., 2014, etc.) various parameters that correlate with the evolution of a fracture system with increasing deformation can be discussed. In this contribution, we test the spacing distribution for normal (Gaussian), log-normal and negative exponential distributions.

The maximum-likelihood estimate (MLE) of parameters of these distributions can be obtained directly in closed form from sample statistics. For instance, the mean spacing $\mu_S$ and standard deviation $\sigma_S$ of a normal distribution are directly given by the sample mean and standard deviation. However, MLE just provides an optimal estimate of the distribution parameters, under the assumption that we have chosen the right distribution, but does not allow comparing which distribution (i.e. which statistical model) better conforms to the data.

The Kolmogorov-Smirnov (K-S) test is a non-parametric test that was developed to check if two samples come from the same distribution, either empirical or theoretical, and it can be used as a goodness-of-fit (GOF) test for the null hypothesis that the empirical distribution function (EDF) is not different from the model cumulative distribution function (CFD). This test is used by many Authors, but it is biased if the parameters of the theoretical distribution are not fixed, but estimated from the data themselves (Wasserman, 2004). For this reason when possible we apply, instead of K-S, the Lilliefors test (Lilliefors, 1967; Lilliefors, 1969) that is not subject to this bias, but unfortunately the Lilliefors test is not available for log-normal distributions. In both cases the results can be expressed as $p - values$ and the null hypothesis of no difference with a given distribution is rejected at 5% significance if $p - value < 0.05$.

In our workflow, if only one type of distribution passes the test, we retain this as the best-fit one. If more than one distribution is retained, and the $p - values$ are similar, we generally discuss the possibility that the data are equally well fitted by both distributions, and that they represent some sort of transitional regime.

## 4 Case studies

### 4.1 Bed-confined joints in turbiditic sandstones

The scanline studied in this case comes from a single turbiditic sandstone bed in the Langhian to Tortonian Marnoso-Arenacea Formation (Val Santerno, Northern Apennines of Italy; Ogata et al., 2017). The scanline was collected on a photogrammetric Digital Outcrop Model (e.g. Bistacchi et al., 2015) with a 5 mm/pixel resolution. The sandstone bed has a variable thickness decreasing from ca. 32 cm to ca. 16 cm over a distance of ca. 95 m. Fractures are bed-confined tensional joints and the hypothesis that we are testing is whether the bed thickness controls joint spacing (as predicted by e.g. Bai and Pollard, 2000).

In Figure 4a the barcode plot shows the position $X_i^F$ of every joint (482 joints in total), with Terzaghi correction already applied.
Here we already see – qualitatively – that fracture density increases from left to right. The trend test based on the Spearman's correlation coefficient yields $R_S^{TREND} = -0.335$ and a very small $p - value \approx 10^{-14}$, hence the null hypothesis of no trend is strongly rejected at a significance level nearing 100%. A further graphical confirmation of this behaviour comes from the CSF (Figure 4a) that shows a continuous curvature indicating that spacing is larger than mean spacing (lower slope) to the left and smaller to the right (higher slope). The same is confirmed by the CSD (Figure 4a), where we notice that the data CSD is lower than the reference CSD in the $0 - 46m$ segment, very close to the reference between $46m$ and $56m$ and then higher than reference up to the end.

Based on these results, we cannot calculate any meaningful statistics on this scanline, since for instance the average fracture spacing is not stationary. The option to subset the scanline to obtain stationary subsets is not feasible, since the trend appears very continuous in the CSD. If we go back to the geological model to be tested – fracturing is controlled by bed thickness – it is natural to try and normalize the data with bed height.

Bed height $H_i^F$ measured at every fracture is shown in Figure 4b. This measurement is affected by some noise (orange line), so we take a smoothed version of this variable, obtained by fitting a polynomial to $H_i^F$ (purple line), as representative of the real bed thickness $H_i^{F-S}$. The normalized fracture position is then calculated as $X_i^{NORM} = X_i / H_i^{F-S}$. In this way the total length of the scanline is altered, but this is not a real issue since we can always switch back to the original reference by using the sequence of fractures along the scanline. Different polynomials of degree between 1st and 10th were tested, and the 2nd degree was chosen as the one that is more successful (i.e. yielding the smaller residuals) in removing the trend. After normalization, the trend test based on the Spearman's correlation coefficient yields $R_S^{TREND} = 0.011$ and $p - value = 0.806$, hence the null hypothesis of no trend for the normalized dataset is retained at a significance level higher than 80%. We see the normalized barcode plot, the CSF and CSF in Figure 4c.

The second stage of analysis shows that normalized spacing is log-normal distributed (Figure 4c), with a $p - value = 0.185$ obtained from the K-S test (the Lilliefors test cannot be performed for this distribution). Exponential and normal distributions are rejected by both the K-S and Lilliefors tests with very low $p - value < 10^{-3}$ (Figure 4c). To confirm the rejection of a random spatial distribution, the $\chi^2$ test performed on the Poisson distribution resulted in a rejection with $p - value \approx 10^{-6}$ (Figure 4c).

From this statistics we conclude that fracture spacing in this turbiditic sandstone layer is controlled by bed thickness and that the joint system is tending towards joint saturation (see discussion in Bai and Pollard, 2000.

**(a) Original scanline dataset**

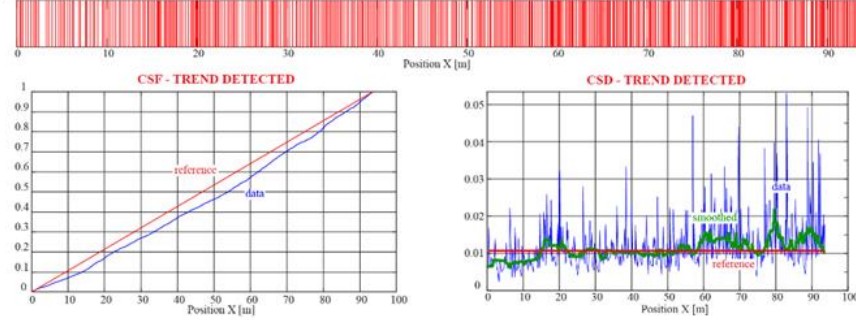

**(b) Bed thickness**

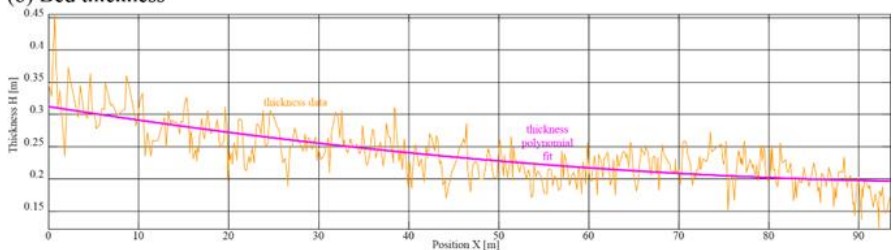

**(c) Thickness-normalized scanline dataset**

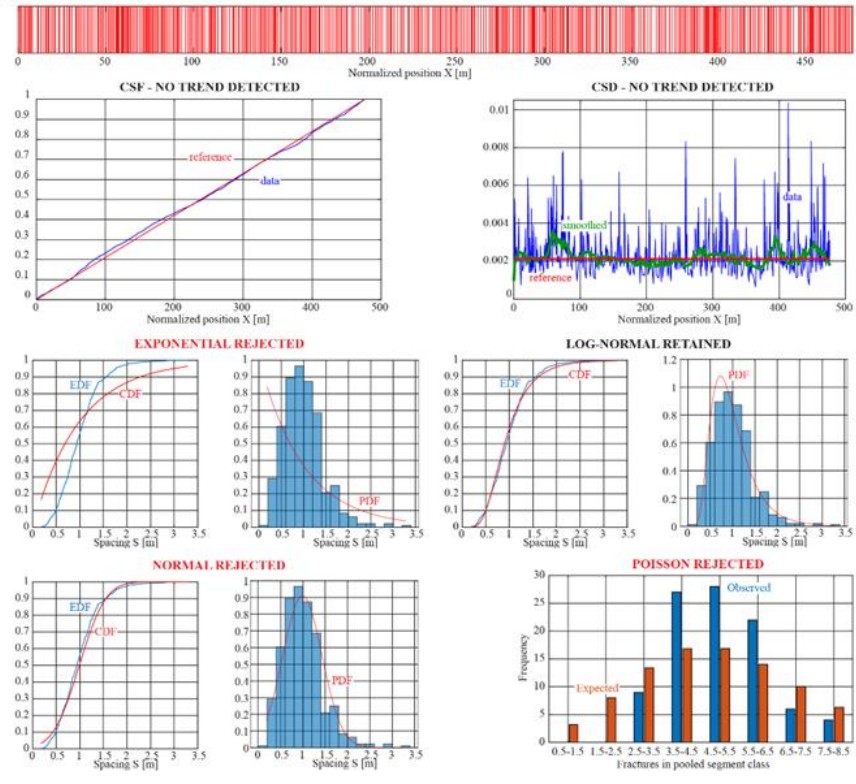

**Figure 4 Analysis for the case study on bed-confined joints in turbiditic sandstones. See discussion in main text.**

## 4.2 Fault-related fracturing in platform carbonates

The scanline discussed in Figure 5 has been collected in the south-eastern part of the Island of Pag (Croatia). In this region, cretaceous carbonate platform successions are folded and imbricated by NW-SE trending thrusts of the External Dinarides
(Korbar, 2009; Mittempergher et al., 2019). The scanline has been collected along a mudstone- wackestone bed in a high-angle anticline forelimb, between two minor subvertical dextral E-W-trending strike-slip faults, with horizontal offsets of 10.3 and 0.5 m respectively. Here we consider as fractures both subvertical joints and extensional veins, developed during the same deformation event (i.e. cogenetic), trending about E-W, parallel to the main faults. Most joints are several meters long and crosscut multiple limestone strata.

The barcode plot (Figure 5a) highlights that, as expected, fracture spacing increases with distance from the fault with the highest offset, i.e. left to right. The Spearman's correlation coefficient test for large-scale trend yields $R_S^{TREND} = 0.438$ and a very small $p - value \approx 10^{-16}$ (Figure 5a), thus the null hypothesis of no trend is strongly rejected at a significance level nearing 100%. The Spearman's correlation coefficient test for local spacing correlation yields $R_S^{LOCAL} = 0.328$ and a $p - value \approx 10^{-9}$ (Figure 5a), hence also the null hypothesis of no pattern/clustering is strongly rejected at a significance level
nearing 100%. The sample CSF and CSD for the whole scanline (Figure 5a) depart significantly from the reference CSF and CSD, with the first part of the function $(1.0 - 5.0\ m)$ having a slope higher than average, the central part $(5.0 - 12.8\ m)$ having a slope close to average, and the final part having a slope lower than average $(12.8 - 19.7\ m)$.

As in the previous example, fracturing is not stationary along this scanline and calculating parameters such as average spacing or fracture density over the entire scanline would have been meaningless. We therefore test the hypothesis that the fault damage
zone can be subdivided in internally homogeneous sectors. These should have relatively constant slopes in the CSF, and we selected their boundaries by recognizing plateaus in the CSD, separated by sharp transitions at 5.0 and 12.8 m (Figure 5a).

When segmented, the three sectors do not show trends or patterns (non-parametric tests for large-scale and local spacing correlation, Figure 5b), and have instead a CSF and CSD compatible with a random or uniform distribution (Figure 5b). We conclude that fractures behave as stationary in the three sectors, hence we can calculate sample statistics such as mean spacing
and its inverse - fracture density (Figure 5d). This suggests that, moving away from the largest fault, fracturing does not decrease continuously, but stepwise.

(a) Original scanline dataset & stationary sectors definition

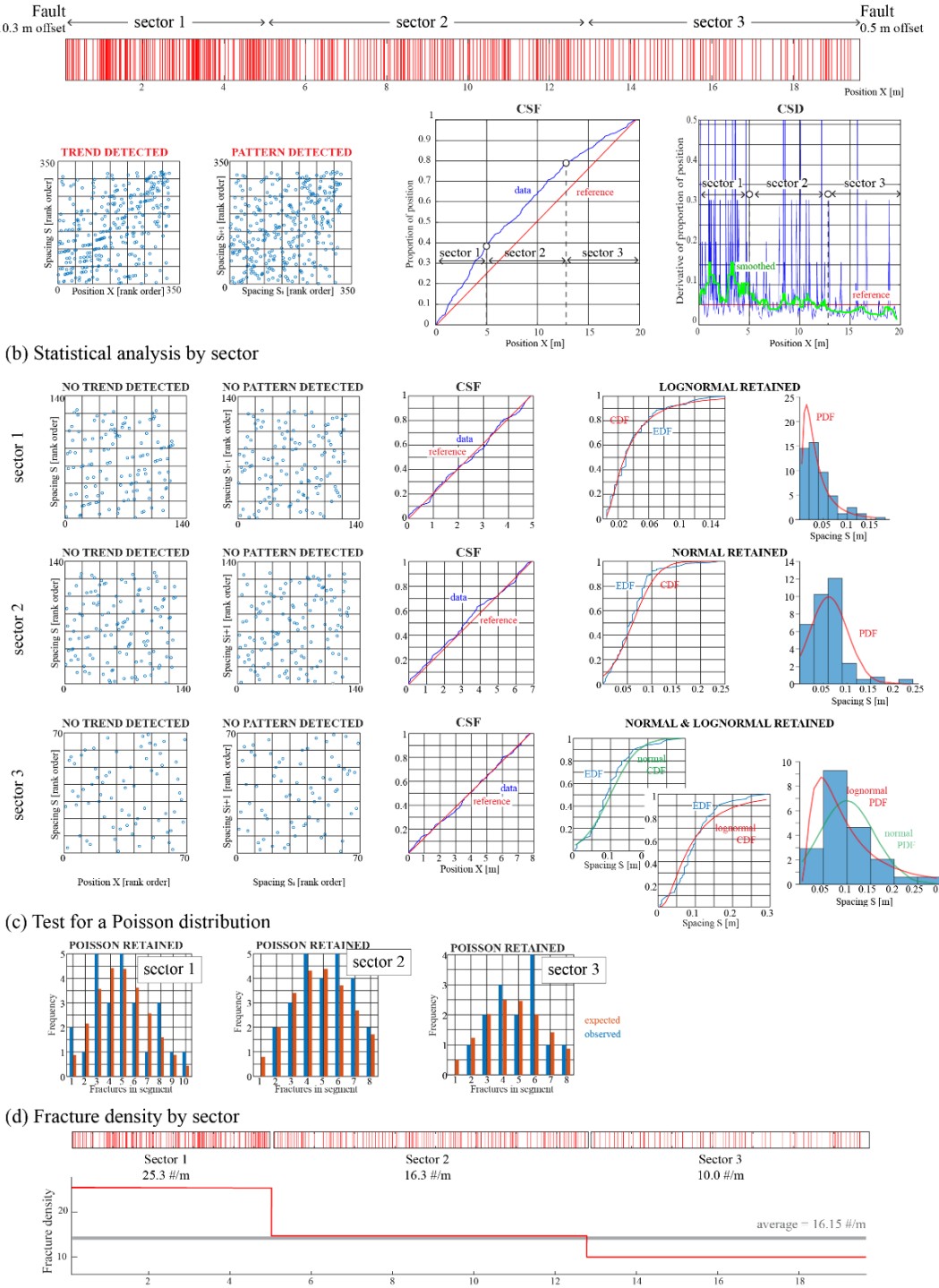

(b) Statistical analysis by sector

(c) Test for a Poisson distribution

(d) Fracture density by sector

**Figure 5 Analysis for the case study on fault-related fracturing in platform carbonates. See discussion in main text.**

Within each stationary sector, a parametric distribution can be fitted to the data. According to the Lilliefors test, no sector has an empiric EDF compatible with an exponential CDF (with significance nearing 100%). Sectors 1 is consistent with a log-normal distribution, sector 2 is compatible with a normal distribution, and sector 3 is compatible with both log-normal and normal distributions. Log-normal and normal distributions indicate a certain degree of organization and fracture saturation, i.e., an evolution towards regularly spaced fractures and the effect of some repulsive process as discussed in the introduction. In this case, where fractures crosscut several mechanical layers and are genetically related with faults, the factors controlling fracture spatial distribution are likely related with inhomogeneous strain distribution (e.g. Spyropoulos et al., 1999).

If we consider the $\chi^2$ goodness-of-fit (GOF) test, we see that fractures in each sector show a spatial distribution apparently compatible with a random Poisson distribution ($p-values = 0.50 \div 0.83$ Figure 5c). This should have implied an exponential spacing distribution, that is strongly excluded by the Lilliefors test. We conclude that the $\chi^2$ GOF test for the Poisson distribution is not reliable in this case, and we confirm the regular spatial distributions implied by normal and log-normal spacing distributions.

**5 Discussion and conclusion**

We have introduced an innovative workflow for the statistical analysis of fracture data collected along scanlines. The workflow is composed of two major stages, each one with alternative options for the analysis of data showing different behaviour and spatial arrangement. Navigating across different options is based on geological hypotheses on the data, that can be validated or falsified by quantitative statistical tests, hence the workflow is both adherent to the geological goals of the analysis and objective under the statistical point of view.

A prerequisite in our analysis is the assessment of stationarity of the dataset. From the statistical point of view, this is motivated by the fact that calculating statistics on non-stationary samples can be meaningless, particularly if the goal is to estimate parameters of the underlying population. From the geological point of view, the normalization and/or sub-setting required to obtain stationary segments of scanline improve our understanding of the deformation processes very much.

For instance, in the first case study normalizing the fracture spacing distribution by bed thickness and observing that the residuals show an almost-saturated log-normal distribution yields an extremely strong demonstration of the effect of layering on jointing (*sensu* Bai and Pollard, 2000), even stronger than in many other published examples where bed thickness is constant.

In the second case study, sub-setting the scanline to obtain stationary and homogeneous domains improved the understanding of fracturing in a damage zone, quite like in Choi et al. (2016) and in Martinelli et al. (2020). Our methodology based on the analysis of CSF and CSD allows setting the boundaries of stationary domains in an objective way, and repeating the non-parametric tests for large-scale and local spacing correlation on the sub-scanlines allows to confirm that the subsets are really

stationary. If applied routinely, this will help shading new light on the internal structure of fault zones (i.e. fault zone architecture studies) and probably in situations where there is a lithological control on brittle deformation.

Once the stationarity requirement has been demonstrated for the data (sub-)set, many statistical methods already known in literature can be applied. In this contribution we discussed mainly methods aimed at understanding the degree of saturation of fracturing based on the type of spacing distribution (i.e. exponential, log-normal, or normal; e.g. Rives et al., 1992; Tan et al., 2014). This is a simple achievement but we would like to recall that it is only possible to use this approach once stationarity has been established, otherwise errors cannot be avoided since the relationship between spatial distribution and spacing

distribution is *not biunivocal*. For instance, it is possible to observe an exponential spacing distribution in case of spacing distributions with a trend (Figure 1e) or repeating pattern (Figure 1d), hence observing an exponential distribution of spacing is not a valid proof of a random spatial distribution. By the way, this impedes using simplified approaches, such as that of the correlation coefficient by Gillespie et al. (1999), without double checking the spatial distribution with other approaches.

On the other hand, advanced analyses such as the normalized correlation count (NCC) by Marrett et al. (2018) can be performed

on stationary datasets defined as proposed here, and will improve the understanding of fracturing in many interesting case studies. If using NCC, it must be evaluated if the non-parametric test for local spacing correlation is really necessary, but we feel that demonstrating large-scale stationarity is still fundamental, at least to be able to use sample distributions to model the underlying population distributions.

Finally, we have also tested a goodness-of-fit test aimed at directly detecting the random Poisson spatial distribution. This

seemed a logical choice since the discrete Poisson distribution is the model of every random distribution of discrete events in 1D, 2D, and 3D, and Poisson processes are used to distribute fractures in Discrete Fracture Network (DFN) simulations (e.g. Dershowitz et al., 2003; Elmo and Stead, 2010; Bonneau et al., 2016). Unfortunately, the $\chi^2$ GOF test seems to return a lot of false negatives, in the sense that fails to reject distributions that are not random, but for instance log-normal (see second case study). For this reason, we prefer to follow the approach presented above instead of using the $\chi^2$ GOF test.

**Acknowledgements**

We warmly acknowledge Angelo Borsani and Andrea Succo who, in addition to collaborating in field data collection, shared many fruitful and happy days in the field with us!

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
