# Peer review of "On a new robust workflow for the statistical and spatial analysis of fracture data collected with scanlines (or the importance of stationarity)"

_Solid Earth, 2020_

## Referee Comment (RC1) · 17 Jul 2020

Comments to author: Manuscript number: se-2020-83 Title: On a new robust workflow for the statistical and spatial analysis of fracture data collected with scanlines (or the importance of stationarity)

To Bistacchi and others,

Overview

The paper presents a workflow to analyze the spatial organization of fractures in scanlines. The main distinguishing contribution of this paper is the use of stationarity as a prerequisite to calculate statistics. The proposed workflow is composed of two stages. The first stage is based on non-parametric analysis mostly based on cross-plots of rank-ordered position and spacings, to detect trends or patterns. If trends are recognized in the first stage, authors propose a normalization or division of scanline into sub-sets, to obtain stationary statistical properties. The second stage is based on a parametric analysis to define the most appropriate distribution to represent each stationary part of the scanline. The paper is a valuable contribution. They present a useful workflow to assess spatial organization of fractures. The discussion about stationarity in fracture networks is also an important concept, especially for statistical modeling and extrapolation purposes. I enjoyed the reading. The paper is well-written and I could understand the objectives, discussion and results. However, I do have minor revisions that I believe will help to improve comprehension of the readers and increase impact of the publication.

General Comments

1. You should add in your introduction more references related to the subject of spatial organization of fractures. Many important publications touching on this topic were not cited (e.g. Laubach et al 2018, Sanderson and Peacock 2019, Hanke et al 2018). The Journal of Structural Geology has a volume dedicated to this subject (Volume 108) and has plenty of relevant references that were not cited. You should compile some of these recent references in your introduction and expose how your method improves the analysis of spatial organization of fractures among what is already published.

2. Fractal distributions are only discussed in line 59, although they are commonly found in fracture systems. There is also no explanation of why power-laws were not included in the parametric tests. As fractal fracture networks have no characteristic size, I doubt that stationarity can be reached when analyzing these systems with your method. You should include a discussion about why you did not use power-laws in the parametric

distribution and if the stationarity-based workflow would work in fractal networks too.

3. The segmentation of the scanline in "sub-sets" is excellent to separate stationary parts of your data. However, it may affect reliability of this workflow to identify spatial organization of clusters, since each domain is going to be analyzed separately. The advantage of methods that do not separate data into sub-sets, such as NCC, is the possibility to identify patterns in a variety of scales. Invariably if a segmentation is done, less information about the spatial organization will be available to be analyzed.

Specific comments

Line 24 – Twiss and Moores, 2006 is not really a reference for traditional fractured formation analysis, it is a general structural geology book. You should add here more examples of fracture analysis literature. A traditional one, to start, and that you should add here, is Pollard and Aydin 1988 that also talks about advancements in the topic along the last century.

Line 43 - You should add here some more references related to the topic of spatial organization of fractures.

Line 55 – Two other references may be worthy to include here. Hooker and Katz 2015 assessed the impact of syn-kinematic cementation on the spatial organization. Cladouhos and Marret 1996 assessed the evolution of fault systems through linkage and interaction during growth.

Line 58:59 – Clustering is also thought to be caused by difference in subcritical crack indexes. Suggest to include Olson's publications in the references and discussion. (e.g. Olson 2004).

Line 61 – What is the scale of Large-scale trends? Please specify.

Line 92 – Another advantage of NCC is that is analyses non-neighbor fracture spacings, making it possible to calculate, for example, cluster spacing and width in the same analysis.

Line 94 – You should explain in more detail what the term "cluster barycentres" mean.

Line 103 – I do not think that stationarity is prerequisite to detect a meaningful statistic. Stationarity should be a prerequisite to extrapolate statistics on a spatial domain (e.g. for geostatistical modeling of spatially distributed variables). Indeed, most used geostatistical algorithms demand stationary parameters as input (e.g. Sequential Gaussian Simulation; Sequential indicator simulation). However, if extrapolation/modeling is not the objective of the statistical analysis, the values are providing reliable information of the sampled scanline, and could be used for other objectives such as comparison between sites (which scanline has the biggest intensity? or which scanline has the biggest CV?). The term "meaningful statistic" is present along the text in many places. I suggest you provide a wider context and explain such impactful statement.

Line 140:143 – Here you should include a discussion about stationarity in fracture networks with fractal or power-law distributed spacings. These types of network may never reach stationarity because they lack characteristic size.

Line 142 - What is the scale of small-scale? Please specify.

Line 200 – The weblink requires a password, could not access it.

Line 221 – It is confusing how you refer to the tests here. Could you be more specific? Maybe you could refer to it as Position-Spacing test and Spacing-Spacing test. Other option is to define clearly in text what are you calling of first and second tests.

Line 309:310 – Interesting conclusion. I imagine that the polynomial function you choose to model the trend is going to impact the resulting residual, i.e. the normalized scanline. It is not clear to me if your method is sensible to overcome poor trend fitting. Probably for each different trend fit, residuals are going to be different too, and that may ultimately affect the type of distribution you identify in the residual parametric test. Did you assess how the residuals changed for each fit using different order polynomials? This assessment may give you and your readers more confidence in the

proposed workflow. It would show that the log-normal distribution identified here is the fracture system signal and not an artifact from the trend removal.

Line 316 - Is there an explanation of why joint and extensional veins were considered together in your analysis? Do you think they pertain to the same fracture set and are genetically related? I imagine joints have no cement in it, and veins have been cemented. Why would joints not be affected by the cementation phase of the veins, if they are timely related? Your analysis of spatial organization considers both joints and veins together. Therefore, they must be interpreted as genetically related. Otherwise the interpretations in lines 344-349 will not be true. Keep in mind that if you are describing the spatial organization of two different sets together, you should not use that information to think about the fracture genesis or evolution, as they could have developed in very different time with different history.

Line 344:349 - That is a very interesting conclusion. My only concern is on the fact that joints and veins are here being analyzed together. If they pertain to the same set, why does joints not been affected by cementation as the veins were? Maybe they have a different timing. If so, I would recommend doing analysis separately. Cementation can affect the spatial organization of fractures (Hooker and katz, 2015; Hooker et al 2018, Laubach et al 2019). An analysis separating them in two sets (in case it is applicable) may yield a different conclusion.

Line 353 – Why is Poisson distribution not reliable? Please explain.

Line 362 - It can be meaningless. I would add that it can be meaningless to extrapolate statistical behavior in a spatial domain.

Line 375:383 – You should add here a discussion of your workflow applied to power-law distributed (fractal) networks. I believe you should also assess power-law distribution in the parametric tests. If your method is not applicable to fractal fracture networks, you should acknowledge here. If it is applicable, explain why you did not use it in the parametric test, since it is a common distribution observed in fracture networks.

[Figure]

Line 386:387 - NCC can identify clusters or periodic behavior even without the normalization by the large-scale trend. It happens because it uses non-neighbor fracture spacings. It may be possible for example to see a periodic behavior in a large scale (10-100m) and a clustered behavior in smaller scales (<1m). If you are using NCC, there is no need to build a stationary data. However, if you use stationary data subsets you will have a partial result, it will not be able to see spatial organization of clusters with respect to each other or spatial organization of large length-scales in the data, because it is segmented.

Technical corrections

Line 291 – blue line color description does not match the figure.

Line 292 – green line color description does not match the figure.

Line 385 – Please rewrite this sentence "...and will help better understanding the development of fracturing...". It does not sound complete.

The papers below are referenced in the text but are not listed in the reference list:

Line 134 – Borradaile, 2003

Line 135 – Caine and Forster, 1999; Mitchel and Faulkner 2009

Line 145 – Wasserman, 2004

Line 266 – Lilliefors, 1967; Lilliefors, 1969

Line 275 – Ogata et al 2017

Line 276 – Bistacchi et al 2015

Line 314 – Korbar, 2009; Mittempergher et al 2019

Line 390 – Dershowitz et al 2003; Elmo and Stead 2010; Bonneau et al 2016

Figure comments

Figure 1/Line 69 – Numbers (1)...(8) are not shown in the figure. Add number in the figure for better comprehension of figure captions.

Figure 1/Line 70 – CSF and CSD were not explained in the text before this figure. I suggest to include the full name of those here, or include a brief explanation in the text before you reference this figure.

Figure 1/Line 72 - The abbreviation EDF was not defined anywhere on text. Please define it somewhere before the figure or in the figure caption.

Figure 5a – The CSD graph shows a blue curve without identification.

Good luck,

---

## Referee Comment (RC2) · Anonymous Referee #2 · 19 Jul 2020

With this paper, authors want to propose a workflow to analyze organization of fracture network using scanlines. This paper is an interesting contribution to weight the different parameters usable to describe a fracture network. This can give in term, an interesting tool to analysis SL on outcrop. Authors highlight their workflow are not applicable for fractal network, Classique observed for fracture network, this assumption must be discussion. To reinforce the demonstration, geological data must be given;

General comments

[Figure]

In the introductive part, several basic and important publications are cited, please add them, see por example the JSG special pub (108) on the topic. A large set of publications deals with the fracture set characterization, could you add them and discuss what is your added value.

You analyze works on 1D from photogrammetry data could you discuss the opportunity to do that in other direction and doing jointed multi-1D analysis?

Line 200 the URL link is locked by a password!! Not usable

First example bed-controlled fractures

Line 275 a view of the outcrop will be helpful Line 275 are data collected automatically are handy made. Fig 4 the thickness of the bed very sharply, why? What is the process to acquire the thickness? What is the error? Lines 283-285 and figure 4 if I agree with your analyze for the 0-56 m section, for the section several break values are observable at 75 and 85 m could you discuss that, and explain why do not you take them into account? Indeed, sections could be described with local spacing correlation, please could you discuss this point? Could you give a stereo-plot of the data? I don't if the data are available, but if you did a Terzaghi correction you may have these data.

Second example fault-controlled fractures

Could you give a view of the outcrop? Could you give a stereo-plot of the data? I don't if the data are available, but if you did a Terzaghi correction you may have these data. And especially for each segment. Are you sure that all the structures are acquired under the same stress field? Here a carefully discussion must develop because this section is a complex interactive structure with the damage zone of two faults probably overlapping, and the change of correlation law could be dependent on the offset of the faults. If I could be agreed with a segmentation of a damage zone, several decrease laws must be tested to valid your hypothesis for the total fracture set. And probably two clusters of fracture are in the sections 1, please discuss a more detailed analysis of

your data, information are here.

What is the representativeness of your curve-fits, what are the residual fractions?

This paper suffers from several form problems and needs careful proofreading

Examples: Line 9 number of the team is not correct Line 31, 98, authors Several ref are forgotten in the bibliography : Borradaile, Caine and Foster, Mitchell and Faulkner, Wasserman, Davis, Ogata, Korbar, Mitteenpergher . . .

Line 285 and others m (it) -> m

On the figures several lines and axis have not legend.
* * *

---

## Editor Comment (EC1) · Roger Soliva (Editor) · 27 Jul 2020

Dear authors,

We now have the feedback from the two reviewers. I ask you to consider and answer to each point mentioned by both reviewer and revise the manuscript accordingly. Many thanks in advance for this work which should improve the quality of the paper.

Best regards, Roger Soliva

---

## Author Comment (AC1) · 3 Oct 2020

Dear Editor, thanks very much to you and to the reviewers for your very useful comments!

We have completed the corrections and we submit the reviewed version of our manuscript se-2020-83 titled "On a new robust workflow for the statistical and spatial analysis of fracture data collected with scanlines (or the importance of stationarity)".

In the attached PDF (se-2020-83-Author_comment.pdf), you will find your letter, the associate editor comments (all in black) and our answers (in blue). As you will see, we have followed the suggestions of both the two reviewers, which agree in the general lines. The comments made by the first reviewer, that were more detailed, are answered in a correspondingly detailed way. The comments made by the second reviewer are answered in detail when they differ from the first reviewer's one, and a reference to the first reviewer's comment appears when the two reviewers basically expressed the same opinion.

Please note that in the re-submitted manuscript file, corrections are highlighted in red, and that just Figure 5 was resubmitted (the others were OK).

With kind regards,

Andrea Bistacchi, Università degli Studi di Milano Bicocca
* * *
Editor's comments

Dear authors,

We now have the feedback from the two reviewers. I ask you to consider and answer to each point mentioned by both reviewer and revise the manuscript accordingly. Many thanks in advance for this work which should improve the quality of the paper.

Best regards, Roger Soliva

We would like to make a general point here. Many comments by both reviewers, and particularly by the first one (Rodrigo Correa), are focused on a comparison between the methodology proposed in our manuscript and the Normalized Correlation Count (NCC) method proposed by Marrett, Gale, Gomez, & Laubach 2018 (JSG, vol. 108). With respect to this comparison, we would like to recall that, as we wrote in the manuscript, our methodology is not intended to be an exclusive alternative to NCC (and the manuscript does not discuss which one is "better" or "worse"). On the other hand, our new methodology (i) could be complementary to NCC in some cases (some of them discussed in the manuscript), (ii) has the advantage of being based on non-parametric statistical methods (so no hypotheses on the shape of statistical distributions is needed), and (iii) has the dis-advantage of not considering (at the moment) fractal power-law distributions (and other possible distributions as well, that are not considered also by NCC). We tried to make these points even more clear in the revised version of the manuscript.
* * *
Reviewr's 1 comments (Rodrigo Correa)

Comments to author: Manuscript number: se-2020-83 Title: On a new robust workflow for the statistical and spatial analysis of fracture data collected with scanlines (or the importance of stationarity)

To Bistacchi and others,

Overview

The paper presents a workflow to analyze the spatial organization of fractures in scanlines. The main distinguishing contribution of this paper is the use of stationarity as a prerequisite to calculate statistics. The proposed workflow is composed of two stages. The first stage is based on non-parametric analysis mostly based on cross-plots of rank-ordered position and spacings, to detect trends or patterns. If trends are recognized in the first stage, authors propose a normalization or division of scanline into sub-sets, to obtain stationary statistical properties. The second stage is based on a parametric analysis to define the most appropriate distribution to represent each stationary part of the scanline. The paper is a valuable contribution. They present a useful workflow to assess spatial organization of fractures. The discussion about stationarity in fracture networks is also an important concept, especially for statistical modeling and extrapolation purposes. I enjoyed the reading. The paper is well-written and I could understand the objectives, discussion and results. However, I do have minor revisions that I believe will help to improve comprehension of the readers and increase impact of the publication.

General Comments

1. You should add in your introduction more references related to the subject of spatial organization of fractures. Many important publications touching on this topic were not cited (e.g. Laubach et al 2018, Sanderson and Peacock 2019, Hanke et al 2018). The Journal of Structural Geology has a volume dedicated to this subject (Volume 108) and has plenty of relevant references that were not cited. You should compile some of these recent references in your introduction and expose how your method improves the analysis of spatial organization of fractures among what is already published.

A reference to the paper by Laubach, Lamarche, Gauthier, Dunne & Sanderson (2018, JSG), that is a general introduction to the cited JSG special volume, has been added to the introduction when introducing spatial distributions (line 44).We have added a reference to Sanderson & Peacock (2019, JSG) when introducing the CSF plot (line 175). However, the paper by Hanke, Fischer & Pollyea (2018, JSG) deals with the semivariogram study of fractures in 2D maps, so we do not feel it would be appropriate to cite it here, in a 1D context.

2. Fractal distributions are only discussed in line 59, although they are commonly found in fracture systems. There is also no explanation of why power-laws were not included in the parametric tests. As fractal fracture networks have no characteristic size, I doubt that stationarity can be reached when analyzing these systems with your method. You should include a discussion about why you did not use power-laws in the parametric distribution and if the stationarity-based workflow would work in fractal networks too.

This comment includes two distinct topics, (i) the very definition of a stationary stochastic process, and (ii) more specifically "fractal" power-law distributions.

Regarding the first one, we feel that there is a misconception here about the definition of a stationary stochastic process. We have better defined this concept in the manuscript at lines 132-135. Regarding

variables showing power-law distributions, they can be indeed stationary if the power-law parameters (exponent and pre-factor) do not change under a transformation (e.g. shift, shrinkage, enlargement) of the spatial domain of the analysis.

More specifically, regarding fractal power-law spacing distributions, we still have not included them in our analysis for two reasons: (1) they are already covered very well by the NCC analysis cited by the reviewer, and (2) we cannot include all possible distributions in this manuscript. For instance, it would have been interesting to test also Pareto distributions, and other ones, but testing all possible distributions exceeds the scope of our present manuscript.

3. The segmentation of the scanline in "sub-sets" is excellent to separate stationary parts of your data. However, it may affect reliability of this workflow to identify spatial organization of clusters, since each domain is going to be analyzed separately. The advantage of methods that do not separate data into sub-sets, such as NCC, is the possibility to identify patterns in a variety of scales. Invariably if a segmentation is done, less information about the spatial organization will be available to be analyzed.

Performing our analysis do not rule out the possibility to perform also NCC. On the contrary, we believe that performing BOTH analyses could be very interesting on some kind of datasets. In any case we feel that our workflow, based on non-parametric statistics, could be very effective particularly to steer the first stages of the analysis.

Specific comments

Line 24 – Twiss and Moores, 2006 is not really a reference for traditional fractured formation analysis, it is a general structural geology book. You should add here more examples of fracture analysis literature. A traditional one, to start, and that you should add here, is Pollard and Aydin 1988 that also talks about advancements in the topic along the last century.

We added Pollard and Aydin (1988), but we like to keep also Twiss and Moores (2006), since the intention of this phrase is to say that fracture studies are a very general and traditional topic in structural geology, covered also with specific chapters in textbooks.

Line 43 (now line 44) - You should add here some more references related to the topic of spatial organization of fractures.

Done, we added here the very good introduction by Laubach et al. (2018), also cited above.

Line 55 – Two other references may be worthy to include here. Hooker and Katz 2015 assessed the impact of syn-kinematic cementation on the spatial organization. Cladouhos and Marret 1996 assessed the evolution of fault systems through linkage and interaction during growth.

Actually, Hooker and Katz (2015) hold (with numerical models) that cementation of joints, resulting in veins, can re-establish the original continuity and mechanical properties of the intact rock mass, hence, in their view, veins do not have the impact on spatial organization (or have a lesser impact) that is shown by joints and open fractures in general. This view is debatable since there are many natural examples where veins actually show a well-developed spatial organization. In any case, we do not feel that entering this discussion here would be useful.

Cladouhos and Marrett (1996) discuss distributions of fault LENGTH, a parameter that cannot be measured in 1D scanlines, so we feel that this reference is off-topic.

Line 58:59 – Clustering is also thought to be caused by difference in subcritical crack indexes. Suggest to include Olson's publications in the references and discussion. (e.g. Olson 2004).

Very good suggestion, thanks! We have added this reference (now at lines 61-62).

Line 61 – What is the scale of Large-scale trends? Please specify.

This was already clarified in the next phrase: the scale of a fault damage zone or of a fold. We rephrased to make this more clear (now line 64).

Line 92 – Another advantage of NCC is that is analyses non-neighbor fracture spacings, making it possible to calculate, for example, cluster spacing and width in the same analysis.

Line 94 – You should explain in more detail what the term "cluster barycentres" mean.

These two comments are related. By "cluster barycentres" we simply mean the weighted centre of each cluster, and the spacing of cluster barycentres is thus the spacing between clusters. We rephrased and added a reference to make this more clear (now at lines 88-89).

Line 103 – I do not think that stationarity is prerequisite to detect a meaningful statistic. Stationarity should be a prerequisite to extrapolate statistics on a spatial domain (e.g. for geostatistical modeling of spatially distributed variables). Indeed, most used geostatistical algorithms demand stationary parameters as input (e.g. Sequential Gaussian Simulation; Sequential indicator simulation). However, if extrapolation/modeling is not the objective of the statistical analysis, the values are providing reliable information of the sampled scanline, and could be used for other objectives such as comparison between sites (which scanline has the biggest intensity? or which scanline has the biggest CV?). The term "meaningful statistic" is present along the text in many places. I suggest you provide a wider context and explain such impactful statement.

This is discussed above in the general comments. Here we slightly rephrased to make it clear that any statistical property characterized over a sampling domain where the property is not stationary, will not tell very much about the underlying population, so it will not be very interesting (now lines 107-108).

Line 140:143 – Here you should include a discussion about stationarity in fracture networks with fractal or power-law distributed spacings. These types of network may never reach stationarity because they lack characteristic size.

This is a misconception already discussed in the general comments above. We have added a general definition of stationarity that should make this clear.

Line 142 - What is the scale of small-scale? Please specify.

Done, thanks (now line 150).

Line 200 – The weblink requires a password, could not access it.

Unfortunately, due to safety issues of the Matlab Web App Server (https://it.mathworks.com/help/releases/R2020a/webappserver/ug/securely-deploying-webapps.html), a password protection is needed. We are adding a message on the server, saying how to obtain an account, and we are discussing the possibility to make the source code available on a public repository (but for this we are waiting for some details on copyright from our IP office).

Line 221 – It is confusing how you refer to the tests here. Could you be more specific? Maybe you could refer to it as Position-Spacing test and Spacing-Spacing test. Other option is to define clearly in text what are you calling of first and second tests.

Ok, good solution, thanks (now line 229-230).

Line 309:310 – Interesting conclusion. I imagine that the polynomial function you choose to model the trend is going to impact the resulting residual, i.e. the normalized scanline. It is not clear to me if your method is sensible to overcome poor trend fitting. Probably for each different trend fit, residuals are going to be different too, and that may ultimately affect the type of distribution you identify in the residual parametric test. Did you assess how the residuals changed for each fit using different order polynomials? This assessment may give you and your readers more confidence in the proposed workflow. It would show that the log-normal distribution identified here is the fracture system signal and not an artifact from the trend removal.

Yes. We have tested different polynomials and we have chosen the one giving the smaller residuals. This was already said at lines 303-304, now slightly rephrased to clarify.

Line 316 - Is there an explanation of why joint and extensional veins were considered together in your analysis? Do you think they pertain to the same fracture set and are genetically related? I imagine joints have no cement in it, and veins have been cemented. Why would joints not be affected by the cementation phase of the veins, if they are timely related? Your analysis of spatial organization considers both joints and veins together. Therefore, they must be interpreted as genetically related. Otherwise the interpretations in lines 344-349 will not be true. Keep in mind that if you are describing the spatial organization of two different sets together, you should not use that information to think about the fracture genesis or evolution, as they could have developed in very different time with different history.

Joints and extensional veins are the same age and belong to the same set. Now this is briefly clarified in the text (line 325-326). We do not indulge in more details since explaining why joints and veins are attributed to the same deformation event in this particular outcrop would not add anything to this methodological manuscript.

Line 344:349 - That is a very interesting conclusion. My only concern is on the fact that joints and veins are here being analyzed together. If they pertain to the same set, why does joints not been affected by cementation as the veins were? Maybe they have a different timing. If so, I would recommend doing analysis separately. Cementation can affect the spatial organization of fractures (Hooker and katz, 2015; Hooker et al 2018, Laubach et al 2019). An analysis separating them in two sets (in case it is applicable) may yield a different conclusion.

See above.

Line 353 – Why is Poisson distribution not reliable? Please explain.

Actually, it is not the distribution but the Goodness of Fit statistical test that is not really powerful. This is clear in our text ("…the $\chi^2$ GOF test for the Poisson distribution is not reliable in this case…").

Line 362 - It can be meaningless. I would add that it can be meaningless to extrapolate statistical behavior in a spatial domain.

Rephrased, thanks (now line 370-371).

Line 375:383 – You should add here a discussion of your workflow applied to power-law distributed (fractal) networks. I believe you should also assess power-law distribution in the parametric tests. If your method is not applicable to fractal fracture networks, you should acknowledge here. If it is applicable, explain why you did not use it in the parametric test, since it is a common distribution observed in fracture networks.

See the general point above. The goal of this manuscript is not to cover all possible distributions.

Line 386:387 - NCC can identify clusters or periodic behavior even without the normalization by the large-scale trend. It happens because it uses non-neighbor fracture spacings. It may be possible for example to see a periodic behavior in a large scale (10-100m) and a clustered behavior in smaller scales (<1m). If you are using NCC, there is no need to build a stationary data. However, if you use stationary data subsets you will have a partial result, it will not be able to see spatial organization of clusters with respect to each other or spatial organization of large length-scales in the data, because it is segmented.

See the general point above. Basically, the idea expressed here is that different analyses must not be seen exclusively, but as different options that can be applied on the same datasets to gain a better insight. In addition, segmenting the datasets might be required for modelling purposes, e.g. to generate DFN networks with properties equivalent to different sectors of a damage zone.

Technical corrections

Line 291 – blue line color description does not match the figure.

Ok, corrected, thanks!

Line 292 – green line color description does not match the figure.

Ok, corrected, thanks!

Line 385 – Please rewrite this sentence "...and will help better understanding the development of fracturing...". It does not sound complete.

Ok rephrased (now line 393), thanks!

The papers below are referenced in the text but are not listed in the reference list:

Line 134 – Borradaile, 2003

Line 135 – Caine and Forster, 1999; Mitchel and Faulkner 2009

Line 145 – Wasserman, 2004

Line 266 – Lilliefors, 1967; Lilliefors, 1969

Line 275 – Ogata et al 2017

Line 276 – Bistacchi et al 2015

Line 314 – Korbar, 2009; Mittempergher et al 2019

Line 390 – Dershowitz et al 2003; Elmo and Stead 2010; Bonneau et al 2016

The Mendeley references were messed up when importing the manuscript to the SE editorial system. We will double check this time. Thanks!

Figure comments

Figure 1/Line 69 – Numbers (1)...(8) are not shown in the figure. Add number in the figure for better comprehension of figure captions.

Done, the numbers were actually misleading, thanks.

Figure 1/Line 70 – CSF and CSD were not explained in the text before this figure. I suggest to include the full name of those here, or include a brief explanation in the text before you reference this figure.

Done, thanks.

Figure 1/Line 72 - The abbreviation EDF was not defined anywhere on text. Please define it somewhere before the figure or in the figure caption.

Done, thanks, see also line 273.

Figure 5a – The CSD graph shows a blue curve without identification.

Fixed, thanks.

Good luck,

Reviewer's 2 comments (Anonymous)

With this paper, authors want to propose a workflow to analyze organization of fracture network using scanlines. This paper is an interesting contribution to weight the different parameters usable to describe a fracture network. This can give in term, an interesting tool to analysis SL on outcrop. Authors highlight their workflow are not applicable for fractal network, Classique observed for fracture network, this assumption must be discussion. To reinforce the demonstration, geological data must be given;

General comments

In the introductive part, several basic and important publications are cited, please add them, see por example the JSG special pub (108) on the topic. A large set of publications deals with the fracture set characterization, could you add them and discuss what is your added value.

Done, see reply to 1st reviewer.

You analyze works on 1D from photogrammetry data could you discuss the opportunity to do that in other direction and doing jointed multi-1D analysis?

This comment is not clear. In 1D, by definition the scanline must be perpendicular to the mean fracture plane, or corrected with Terzaghi correction, and this is what we are doing. Any other linear sampling would return a misleading measure with apparent spacing larger than true spacing.

Line 200 the URL link is locked by a password!! Not usable

See reply to 1st reviewer.

First example bed-controlled fractures

Line 275 a view of the outcrop will be helpful Line 275 are data collected automatically are handy made. Fig 4 the thickness of the bed very sharply, why? What is the process to acquire the thickness? What is the error? Lines 283-285 and figure 4 if I agree with your analyze for the 0-56 m section, for the section several break values are observable at 75 and 85 m could you discuss that, and explain why do not you take them into account? Indeed, sections could be described with local spacing correlation, please could you discuss this point? Could you give a stereo-plot of the data? I don't if the data are available, but if you did a Terzaghi correction you may have these data.

There are typos and a mix of different points in this comment, so not everything is clear. Anyway photos of the outcrops, stereoplots, crosscutting relationships and other structural data, and other geological details as well, are omitted in this methodological manuscript due to space requirements. They are all already published in Ogata et al. (2017, GSA Bull.), cited here.

Second example fault-controlled fractures

Could you give a view of the outcrop? Could you give a stereo-plot of the data? I don't if the data are available, but if you did a Terzaghi correction you may have these data. And especially for each segment. Are you sure that all the structures are acquired under the same stress field? Here a carefully discussion must develop because this section is a complex interactive structure with the damage zone of two faults probably overlapping, and the change of correlation law could be dependent on the offset of the faults. If I could be agreed with a segmentation of a damage zone, several decrease laws must be tested to valid your hypothesis for the total fracture set. And probably two clusters of fracture are in the sections 1, please discuss a more detailed analysis of your data, information are here.

See previous comment.

What is the representativeness of your curve-fits, what are the residual fractions?

See reply to 1st reviewer.

This paper suffers from several form problems and needs careful proofreading

Examples: Line 9 number of the team is not correct Line 31, 98, authors Several ref are forgotten in the bibliography : Borradaile, Caine and Foster, Mitchell and Faulkner, Wasserman, Davis, Ogata, Korbar, Mitteenpergher….

Fixed. See reply to 1st reviewer.

Line 285 and others m (it) -> m

Sorry, this comment is not clear.

On the figures several lines and axis have not legend.

The axis labels are omitted (to make figures less cluttered) just where they are obvious, as for instance when plotting a cumulative distribution function ranging from 0.0 to 1.0.

---

## Author Response (AR2)

Dear Editor, thanks very much for your comments!

Regarding the two points that you ask us to reconsider, I answer here below (adding your points in blue for clarity).

1 - Why you did not distinguished joints from veins ? I fully agree with the R#1 comment. I have some reserves on the fact they are co-genetic. Generally, joints and veins do not form unber the sames processes and stress conditions. What study or clear element support your conclusion ? I also recommend doing analysis separately.

We finally understood the source of the problem that you rise here. In the original manuscript we kept to a minimum the geological and tectonic description of the outcrops used as case studies, since the goal was to discuss the statistical methods, and not tectonic or structural aspects of these particular fracture sets. Particularly for the second case study, we said very quickly that fractures and veins are cogenetic and that for this reason we consider them altogether in the analysis. Actually, the story is a bit more complicated, as we have now explained at lines 329-334 (in blue in the manuscript pdf):

"Closely-spaced subvertical extensional fractures and veins, striking nearly parallel to the faults, crosscut folded bedding planes and are therefore associated with the last stages of fold tightening, or postdate folding. Extensional fractures, which are generally longer than veins, bear patches of blocky calcite having the same appearance of that cementing the veins. This evidence suggests the hypothesis that fractures and veins were cogenetic and both associated with the activity of strike-slip faults. To test the relations between faults, veins and fractures, here we considered extensional fractures and veins as a unique fracture set."

In a few words, what appear now as "veins" and "joints" where simply joints during active deformation, then veins were filled by late-stage or post-tectonic calcite, and the larger ones were not completely filled (but they show "bridges" of the same calcite as in veins), hence they still resemble joints. We also have preliminary isotopic data supporting this hypothesis, but we feel it would be too much to discuss them here in this kind of manuscript, that is mainly aimed at discussing statistical methods.

2 - A minor point. Even if Hooker and Katz (2015) reveal that vein probably have little impact on spatial organisation, they are fractures and present, so they can be counted. I do not mean you have to fully discuss this process, but there is no clear reason to exclude this reference. I also think you could refer to other work with respect to the massive bibliography existing in this topic.

 We have added the Hooker and Katz (2015) reference at line 57-58 (in blue in the manuscript pdf).

Thank you very much!

With kind regards,

Andrea Bistacchi, Università degli Studi di Milano Bicocca

---

## Author Response (AR3)

Dear Editor, thanks very much for the final acceptance of our manuscript! We are uploading the final files, with just a slight change that was already announced during the review process. We are opening (it will be operational tomorrow) a public repository to freely distribute the Matlab® App, with a user-friendly graphical user interface, that we developed to perform this analysis. The code will be available for download on a public repository at https://github.com/bistek/DomStudioFracStat1D, that is referenced as DOI:10.5281/zenodo.4122291. We changed the text accordingly at lines 189-190 and in the acknowledgments.

Thank you very much!

With kind regards,

Andrea Bistacchi, Università degli Studi di Milano - Bicocca